# Wnt/β-Catenin Signaling Regulates Yap/Taz Activity during Embryonic Development in Zebrafish

**DOI:** 10.3390/ijms251810005

**Published:** 2024-09-17

**Authors:** Matteo Astone, Chiara Tesoriero, Marco Schiavone, Nicola Facchinello, Natascia Tiso, Francesco Argenton, Andrea Vettori

**Affiliations:** 1Department of Biology, University of Padua, 35131 Padua, Italy; matteo.asto@gmail.com (M.A.); natascia.tiso@unipd.it (N.T.); 2Department of Biotechnology, University of Verona, 37134 Verona, Italy; chiara.tesoriero@univr.it; 3Department of Molecular and Translational Medicine, University of Brescia, 25123 Brescia, Italy; marco.schiavone@unibs.it; 4Neuroscience Institute, Italian National Research Council (CNR), 35131 Padua, Italy; nicola.facchinello@cnr.it

**Keywords:** Wnt/β-catenin, Yap/Taz, Hippo, zebrafish, crosstalk, embryonic development

## Abstract

Hippo-YAP/TAZ and Wnt/β-catenin signaling pathways, by controlling proliferation, migration, cell fate, stemness, and apoptosis, are crucial regulators of development and tissue homeostasis. We employed zebrafish embryos as a model system to elucidate in living reporter organisms the crosstalk between the two signaling pathways. Co-expression analysis between the Wnt/β-catenin *Tg(7xTCF-Xla.Siam:GFP)^ia4^* and the Hippo-Yap/Taz *Tg(Hsa.CTGF:nlsmCherry)^ia49^* zebrafish reporter lines revealed shared spatiotemporal expression profiles. These patterns were particularly evident in key developmental regions such as the midbrain–hindbrain boundary (MHB), epidermis, muscles, neural tube, notochord, floorplate, and otic vesicle. To investigate the relationship between the Wnt/β-catenin pathway and Hippo-Yap/Taz signaling in vivo, we conducted a series of experiments employing both pharmacological and genetic strategies. Modulation of the Wnt/β-catenin pathway with IWR-1, XAV939, or BIO resulted in a significant regulation of the Yap/Taz reporter signal, highlighting a clear correlation between β-catenin and Yap/Taz activities. Furthermore, genetic perturbation of the Wnt/β-catenin pathway, by APC inhibition or DKK1 upregulation, elicited evident and robust alteration of Yap/Taz activity. These findings revealed the intricate regulatory mechanisms underlying the crosstalk between the Wnt/β-catenin and Hippo-Yap/Taz signaling, shedding light on their roles in orchestrating developmental processes in vivo.

## 1. Introduction

Unveiling the complex network of relationships and interactions between different signaling pathways in the complexity of a tissue, organ, or entire organism is fundamental to clarify how critical biological processes are orchestrated at the cellular and molecular level.

Hippo-YAP/TAZ and Wnt/β-catenin signaling pathways are crucial in maintaining tissue homeostasis and organ size by regulating proliferation, migration, cell fate, stemness, and apoptosis [1,2]. Dysregulation of these pathways alters critical biological processes such as development and regeneration and is also a leading cause of tumorigenesis [1,3].

Hippo-YAP/TAZ signaling is a central regulator of tissue growth, and several upstream signals coordinate its activation, including cell density, cell-cell contact, mechanical force, growth factors, and GPCR signaling [4]. As downstream effectors of MST1/2, (the mammalian orthologs of Hippo), YAP1 (Yes Associated Protein 1), and TAZ (or WWTR1-WW Domain Containing Transcription Regulator 1) act as transcriptional regulators whose function is controlled by their phosphorylation status within the Hippo-Yap/Taz signaling cascade. When Hippo (or MST1/2 in mammals) is inactive, non-phosphorylated YAP1 and TAZ can translocate into the nucleus to drive the transcription of a set of target genes as co-transcription factors of TEADs [5,6]. When Hippo (MST1/2) is active, the sterile 20-like kinases MST1/STK4 and MST2/STK3 phosphorylate and activate LATS1/2 kinases, which in turn phosphorylate YAP1 and TAZ, blocking their nuclear translocation [6,7,8,9,10,11,12,13].

Canonical Wnt signaling relies on the regulation of the transcriptional co-activator β-catenin by a cytoplasmic destruction complex, composed of the central scaffold protein AXIN, the adenomatous polyposis coli (APC), and the two kinases casein kinase 1 (CK1) and glycogen synthase kinase-3 (GSK3). When canonical Wnt signaling is off, CK1 and GSK3 phosphorylate β-catenin, driving it to β-TrCP-mediated ubiquitylation and consequent proteasomal degradation. Wnt ligands bind to Frizzled (Fz) transmembrane receptors and lipoprotein receptor-related protein 5 or 6 (LRP5/6). Conversely, direct binding to LRP5/6 by the secreted Dikkopf (DKK) proteins antagonizes Wnt signaling. The Wnt/Fz/LRP5/6 complex formation enables Fz-mediated interaction with the cytoplasmic protein Dishevelled (Dvl). Dvl and LRP5/6 cytoplasmic tails are phosphorylated by CK1 and GSK3, leading to the recruitment of AXIN from the destruction complex to the plasma membrane. This is the key event causing the functional inactivation of the destruction complex, β-catenin accumulation, and its relative nuclear translocation. Within the nucleus, β-catenin binds TCF/Lef transcription factors, controlling the transcription of Wnt/β-catenin target genes [14].

Wnt signaling emerged as a potent regulator of YAP/TAZ nuclear activity. A significant fraction of Wnt target genes and Wnt-induced biological responses are YAP/TAZ-dependent [15]. Mechanistically, YAP1 and TAZ are integral components of the β-catenin destruction complex, which also serves as a cytoplasmic sink for YAP/TAZ. YAP/TAZ association to the destruction complex is mediated by AXIN. Thus, by promoting LRP5/6-mediated AXIN recruitment at the plasma membrane, Wnt signaling physically dislodges YAP/TAZ from the destruction complex. Consequently, YAP/TAZ can enter the nucleus and activate transcription [16]. Within the destruction complex, phosphorylated β-catenin bridges TAZ to its ubiquitin ligase β-TrCP, leading to TAZ degradation [17].

YAP/TAZ also act as antagonists of Wnt/β-catenin signaling in the cytoplasm since YAP/TAZ associated with AXIN within the destruction complex are essential for β-TrCP recruitment to the complex. Therefore, YAP/TAZ are required for β-catenin degradation in the absence of Wnt signals, while depletion of YAP/TAZ can lead to the activation of β-catenin without the need for a Wnt upstream signal [16]. Moreover, TAZ was shown to inhibit the CK1-mediated phosphorylation of Dvl, thus blocking Wnt signal transduction from the plasma membrane to the destruction complex [18]. Hippo-YAP/TAZ and Wnt/β-catenin pathways are even more transcriptionally intertwined, as YAP1 has been shown to interact with β-catenin. YAP1-TEAD and YAP1-β-catenin complexes influence and modulate each other’s activity, playing roles of different relevance depending on the cell type and context. In some cases, they may even form a complex with TCF (TCF-β-catenin-YAP1-TEAD), which then translocates to the nucleus to regulate the expression of common target genes of TEAD and TCF. This intricate interplay between Hippo-YAP/TAZ and Wnt/β-catenin pathways adds another layer of complexity to their regulatory mechanisms and underscores their importance in various biological processes [19,20,21]. A growing body of evidence suggests that the Wnt pathway interacts with Hippo-YAP/TAZ at the level of rhombomeres boundaries by regulating cell proliferation, cell fate, and tissue homeostasis, thus inducing a proper hindbrain segmentation [16,17,22,23]. Crosstalk between Hippo-YAP/TAZ and Wnt/β-catenin pathways is also evident in the development of other organs, including intestine, where they regulate the cellular turnover in the intestinal crypts [24], heart, where they control cardiomyocytes proliferation and organ size [25,26], kidney, where they have a key role in epithelial tubulogenesis [27], and bones, where they regulate osteoblasts proliferation and differentiation [20].

Active cooperation between the two signaling pathways is observed not only in organ development during embryogenesis but also during disease development, such as gastrointestinal cancers [21], breast cancer [28], kidney cancer and kidney cystic fibrosis [29], and neurodegenerative disorders [30].

Here, we undertake an in vivo analysis of the crosstalk between Hippo-Yap/Taz and Wnt/β-catenin signaling pathways to observe the ability of the Wnt/β-catenin pathway to regulate YAP/TAZ activity during development at the whole organism level. We take advantage of the zebrafish transgenic reporter line *Tg(Hsa.CTGF:nlsmCherry)^ia49^* reporting in a living organism Yap/Taz bona fide transcriptional activity YAP/TAZ [31].

## 2. Results

### 2.1. Wnt/β-Catenin and Hippo-Yap/Taz Pathways Are Characterized by an Overlapping Spatial Expression during Zebrafish Development

A growing body of evidence suggests that the Wnt pathway can interact with Yap/Taz signaling. Several recent in vitro studies showed that Yap/Taz activity can be regulated by the Wnt/β-catenin pathway through sequestration of Yap/Taz in different cell types [16,17,32,33]. However, to date, no data are available to demonstrate the occurrence of this interaction during embryonic development. For this reason, we evaluated whether the Wnt and Hippo-Yap/Taz pathways exhibit an overlapping spatial organization during development in zebrafish embryos. We examined, through confocal microscopy, zebrafish embryos of the *Tg(7xTCF-Xla.Siam:GFP)^ia4^* and *Tg(Hsa.CTGF:nlsmCherry)^ia49^* transgenic lines, reporting in vivo Wnt/β-catenin and Hippo-Yap/Taz activity, respectively. The co-expression analysis revealed shared dynamic spatiotemporal patterns. In fact, the Wnt/β-catenin and Hippo-Yap/Taz pathways exhibit areas of overlapping spatial expression during zebrafish development, although this overlap is not uniform across all tissues. The expression profiles of both transgenes in 48 h post-fertilization (hpf) embryos clearly demonstrated the presence of partial signal colocalization in the midbrain–hindbrain boundary (MHB), muscles, neural tube, notochord, floorplate, and otic vesicle (Figure 1). We used Mander’s Overlap Coefficient (MOC) to quantify the degree of colocalization between fluorophores. The coefficients in different regions consistently exceeded 0.8, with 0 indicating no overlap and values closer to 1 reflecting partial colocalization between the two reporter lines (Figure 1 and Appendix A).

### 2.2. Pharmacological Inhibition of the Wnt/β-Catenin Pathway Reduces Yap/Taz Activity during Zebrafish Embryonic Development

Given the expression of Yap/Taz and Wnt signaling components in different tissues, we aimed to explore the potential synergies between these two pathways. To assess Wnt/β-catenin-mediated regulation of Yap/Taz activity during development, we employed the Yap/Taz zebrafish reporter *Tg(Hsa.CTGF:nlsmCherry)^ia49^* [31] as an indicator of Yap/Taz modulation upon perturbation of the Wnt pathway through a combination of pharmacological and genetic interventions.

To test whether the modulation of Wnt/β-catenin may interfere with the activity of Yap/Taz, we performed different experiments using chemical inhibitors able to target specific components of the Wnt pathway. Specifically, Wnt/β-catenin inhibition was achieved using IWR-1 and XAV939, which increase AXIN levels by inhibiting tankyrases 1 and 2 (TNKS1 and TNKS2) [34]. TNKS1/2 binds directly to AXIN proteins and regulates AXIN levels through poly-ADP-ribosylation and ubiquitylation [35,36,37,38]. Under physiological conditions, AXIN represents the limiting factor in the β-catenin destruction complex [1]. Therefore, AXIN upregulation by IWR-1 or XAV939 increases β-catenin cytoplasmic retention and degradation, negatively impacting Wnt signaling.

We treated the *Tg(Hsa.CTGF:nlsmCherry)^ia49^* Yap/Taz zebrafish reporter line with IWR-1 and XAV939 and then assessed Yap/Taz activity by evaluating the modulation of mCherry fluorescence levels in whole embryos. In embryos exposed to IWR-1 or XAV939 for 24 h, a general and significant decrease in mCherry signal was observed, compared to age-matched control embryos treated with DMSO (Figure 2A,B and Appendix A). This downregulation suggests that inhibition of the β-catenin destruction complex can indeed reduce Yap/Taz nuclear activity in vivo. To independently validate the results obtained with the *Tg(Hsa.CTGF:nlsmCherry)^ia49^* Yap/Taz reporter line, we also assessed by qPCR analysis the expression levels of two Yap/Taz responsive genes (ccn2a and ccn2b) in embryos treated with XAV939. Consistently, we found that the inhibition of Wnt/β-catenin effectively downregulates the expression of ccn2a and ccn2b, as a consequence of reduced Yap/Taz activity (Figure 2C).

### 2.3. Genetic Inhibition of the Wnt/β-Catenin Pathway Reduces Yap/Taz Activity during Zebrafish Embryonic Development

To validate the existence of potential interactions between the Wnt/β-catenin and Hippo-Yap/Taz signaling pathways, we also employed a genetic approach to inhibit Wnt/β-catenin signaling. We utilized the transgenic zebrafish line *Tg(hsp70:dkk1-GFP)^w32^* [39], which is a valuable tool for modulating Wnt/β-catenin activity in zebrafish larvae. This transgenic line expresses Dkk1, an antagonist of the Wnt pathway, fused with Green Fluorescent Protein (GFP), under the control of a heat shock-inducible promoter (hsp70). Heat shock induction triggers the expression of Dkk1-GFP, effectively inhibiting Wnt/β-catenin signaling in a temporally controlled manner. This genetic manipulation allowed us to investigate the consequences of Wnt/β-catenin pathway inhibition on downstream signaling events and its potential influence on Yap/Taz activity.

With this purpose, we outcrossed the *Tg(Hsa.CTGF:nlsmCherry)^ia49^* reporter fish with the *Tg(hsp70:dkk1-GFP)^w32^* line and subjected the offspring to three rounds of heat shock treatment from 12 to 72 hpf. After the ubiquitous induction of Dkk1, we observed a significant reduction in mCherry fluorescence (Figure 3A,B), albeit somewhat less pronounced compared to the results obtained from the pharmacological inhibition experiments. We therefore decided to examine changes in mCherry also at the mRNA level in *Tg(hsp70:dkk1-GFP)^w32^*/*Tg(Hsa.CTGF:nlsmCherry)^ia49^* embryos. This approach allowed us to better appreciate subtle variations of Yap/Taz activity after heat shock treatment because the mRNA turnover is generally faster and thus more responsive compared to the corresponding protein modulation. The whole mount in situ hybridization conducted to detect the mCherry transcripts revealed a pronounced and general reduction in the mCherry mRNA levels in embryos overexpressing Dkk1 compared to the control siblings (Figure 3C–E).

In addition to our findings with the pharmacological inhibition approach, these data reinforce the notion that inhibiting the Wnt/β-catenin signaling pathway—whether at the level of the β-catenin destruction complex (using IWR-1 and XAV939) or at the level of Wnt/receptor interaction (using DKK1)—results in a decrease in Yap/Taz activity in vivo. This convergence of evidence strengthens our understanding of the intricate interplay between Wnt/β-catenin and Hippo-Yap/Taz signaling pathways in developmental processes and tissue homeostasis [5,40].

### 2.4. Pharmacological Inhibition of β-Catenin Kinase GSK3 during Zebrafish Embryonic Development Increases Yap/Taz Activity

Building upon the findings of the Wnt-inhibiting experiments, we investigated whether the Wnt/β-catenin signaling pathway could also positively regulate Yap/Taz activity. For this purpose, we tested on *Tg(Hsa.CTGF:nlsmCherry)^ia49^* embryos the chemical agonist of the Wnt pathway, 6-Bromoindirubin-3′-oxime (BIO), which inhibits the activity of GSK3, the kinase that promotes the β-catenin phosphorylation and its consequent degradation. *Tg(Hsa.CTGF:nlsmCherry)^ia49^* Yap/Taz reporter embryos were treated with BIO from 24 to 48 hpf, and the mCherry expression levels were measured and compared with the transgene expression of the untreated control offsprings. Unfortunately, it was impossible to observe the effect of the Wnt agonist on mCherry expression levels of *Tg(Hsa.CTGF:nlsmCherry)^ia49^* embryos, as the BIO compound itself emits red fluorescence. To overcome this problem, we employed in situ hybridization, which allowed us to analyze the transgene expression in *Tg(Hsa.CTGF:nlsmCherry)^ia49^* embryos using an antisense probe targeting the mCherry transcript. Interestingly, our experiments revealed a significant increase in mCherry mRNA levels in embryos treated with the drug compared to DMSO-treated controls (Figure 4A–C). This finding suggests that β-catenin phosphorylation negatively influences Yap/Taz transcriptional activity in vivo during development, supporting the idea of the existence of a direct interplay between Wnt/β-catenin and Hippo-Yap/Taz signaling pathways.

Wnt-mediated up-regulation of Yap/Taz activity observed in the *Tg(Hsa.CTGF:nlsmCherry)^ia49^* reporter line was validated by qPCR analysis of ccn2a and ccn2b expression levels in embryos treated with BIO. Remarkably, results from this independent experiment corroborated our previous findings, as we observed a significant increase in the expression of ccn2a and ccn2b upon activation of Wnt/β-catenin signaling, indicative of a Yap/Taz activity upregulation (Figure 4D).

### 2.5. Activation of the Wnt/β-Catenin Pathway through Apc Depletion Increases Yap/Taz Activity during Zebrafish Development

To independently confirm the interaction between the Wnt/β-catenin and Hippo-Yap/Taz signaling pathways, we decided to use an alternative strategy, exploiting the genetic repression of the destruction complex induced by the depletion of the APC gene. For this analysis, we crossed the *Tg(Hsa.CTGF:nlsmCherry)^ia49^* transgenic strain with the apc^hu745^ zebrafish mutant line, which harbors a loss of function mutation in the apc gene [41]. The loss of the Apc protein causes the destabilization of the β-catenin destruction complex, with the subsequent release of its components (including β-catenin), which in turn promotes the Wnt activation.

Apc^hu745^ homozygous mutants display an altered phenotype during early development, characterized by the absence of pharyngeal arches at 72 hpf, as demonstrated by alcian blue staining in Figure 5A. Typically, the *Tg(Hsa.CTGF:nlsmCherry)^ia49^* embryos present a robust transgene expression in the pharyngeal arches under physiological conditions. However, in the apc^hu745^ mutant background, the mCherry signal is absent due to the lack of these anatomical structures, as shown in Figure 5B.

For this reason, our analysis was redirected toward other anatomical districts in which the reporter is typically active. Through in situ hybridization, we observed a strong up-regulation of the mCherry transgene expression in the midbrain–hindbrain boundary, eyes, otic vesicle, heart and vascular system in *Tg(Hsa.CTGF:nlsmCherry)^ia49^* apc^hu745^ embryos, as a consequence of the β-catenin stabilization (Figure 5C–E). In summary, these results confirmed once again in vivo that during development Yap/Taz activity is positively regulated by the activation of Wnt signaling pathway mediated by the β-catenin. This consistency across experiments underscores the robustness of the observed relationship between Wnt/β-catenin and Hippo-Yap/Taz signaling pathways, highlighting their interconnected roles during developmental processes.

## 3. Discussion

The Hippo-Yap/Taz pathway is considered an integrator of information from environmental and cell-type-specific signals, local extracellular matrix (ECM) components, and mechanical forces. These factors modulate the subcellular localization of Yap/Taz and alter cell behavior, but the regulation mechanisms of Yap/Taz nucleocytoplasmic shuttling and their implications for cellular homeostasis are still under investigation [42,43,44,45]. In this context, our findings provide in vivo evidence that supports a model where Wnt/β-catenin signaling can indeed modulate YAP/TAZ cellular localization, offering insights into how these pathways can co-regulate important developmental processes in zebrafish embryos.

Interestingly, two previous studies that have evaluated the direct relationship between Wnt and Hippo-YAP/TAZ signaling pathways reported contrasting outcomes. Park and collaborators provided evidence supporting the role of non-canonical Wnt signaling in regulating YAP/TAZ activity. Through a series of in vitro experiments, the authors demonstrated that non-canonical Wnt ligands activate YAP/TAZ through pathways independent of β-catenin. This model suggests that the non-canonical Wnt pathway can activate Rho GTPases, which in turn influence the phosphorylation state of YAP/TAZ, consequently affecting their nuclear localization and transcriptional activity [32].

These findings contrast with a previously proposed model, in which the canonical Wnt signaling directly affects TAZ and YAP activity through the β-catenin destruction complex. In their work, Azzolin and colleagues elegantly demonstrated that components of the destruction complex sequestered upon Wnt activation can prevent the phosphorylation and subsequent degradation of YAP/TAZ. This model places YAP and TAZ as integral components within the canonical Wnt signaling cascade, directly linking Wnt activation to YAP/TAZ-mediated transcriptional activity [16,17].

Interestingly, our in vivo analysis supports the model proposed by Azzolin, as we observed significant alterations in YAP/TAZ activity upon modulation of the canonical Wnt pathway in zebrafish embryos. Our study used the *Tg(Hsa.CTGF:nlsmCherry)^ia49^* zebrafish transgenic reporter as a readout of Yap/Taz activation in vivo during development.

Even though the well-documented genetic redundancy in zebrafish could theoretically impact the interpretation of pathway-specific signaling activities, the use of specific signaling pathway transgenic reporter lines in our study provided significant advantages over other in vitro approaches. The *Tg(Hsa.CTGF:nlsmCherry)^ia49^* line is designed to specifically report the nuclear transcriptional activity of Yap and Taz, the final transducers of the Hippo/Yap/Taz pathway. This design minimizes potential interference from upstream signal redundancies, ensuring that the observed mCherry signals accurately reflect Yap/Taz activity [31]. Similarly, the Wnt/β-catenin reporter line *Tg(7xTCF-Xla.Siam:GFP)^ia4^* utilizes well-characterized β-catenin responsive elements, making it a robust and reliable tool for monitoring Wnt pathway activity in vivo.

Through genetic and pharmacological modulation of the β-catenin destruction complex activity, we observed Yap/Taz reporter signal variations, mirroring changes in β-catenin activity. Increased stability of the destruction complex, induced by IWR-1, XAV939, or Dkk overexpression, led to inhibition of Yap/Taz transcriptional activity. As Yap1 and Taz are integral components of the β-catenin destruction complex, any factor that enhances the stability of this structure increases the concomitant sequestration of both β-catenin and Yap/Taz in the cytoplasm and counteracts the nuclear translocation of these transcriptional regulators. Conversely, we observed that destabilization of the destruction complex increased *Tg(Hsa.CTGF:nlsmCherry)^ia49^* reporter signal, reflecting the enhanced Yap/Taz transcriptional activity.

Under physiological conditions, the Apc protein is part of a destruction complex that helps to regulate the levels of β-catenin by promoting its degradation. However, this destruction complex becomes destabilized when the Apc protein is mutated, as in the apc^hu745^ knockout embryos. As a result, β-catenin is no longer targeted for degradation and is instead released into the cytoplasm, promoting the activation of the Wnt signaling pathway. Therefore, the transgene up-regulation observed in *Tg(Hsa.CTGF:nlsmCherry)^ia49^*/apc^hu745^ embryos supports the notion that Yap1 and Taz are integral components of the β-catenin destruction complex, and alterations of the complex stability impact the nuclear translocation and transcriptional activity of these regulators of the Hippo pathway. Furthermore, the *Tg(Hsa.CTGF:nlsmCherry)^ia49^* Yap/Taz reporter system was also fundamental to investigating the role of β-catenin phosphorylation in driving TAZ degradation during embryonic development. By blocking β-catenin phosphorylation with BIO, we observed an enhanced nuclear translocation of β-catenin, consistent with its escape from the destruction complex. In line with previous research demonstrating that phosphorylated β-catenin facilitates TAZ ubiquitination and degradation (17), our findings in the *Tg(Hsa.CTGF:nlsmCherry)^ia49^* reporter line clearly showed that treatment with BIO promoted the release of TAZ from the complex, enhancing Yap/Taz activity. These results validate the Wnt/β-catenin-mediated regulation of Yap/Taz nuclear activity observed in vitro by Azzolin and colleagues [16], now demonstrated in a living vertebrate model.

Additionally, it is worth noting that the partial overlap observed in certain regions may indicate complex regulatory interactions between these pathways, potentially in a tissue-specific manner. This complexity aligns with findings from other studies, which suggest dynamic and context-dependent crosstalk between these signaling pathways.

The insights gained from our study regarding the Wnt/β-catenin pathway have broader implications, indicating the feasibility of analyzing the interactions between Yap/Taz and other signaling pathways implicated in vertebrate development and cancer. Taking advantage of living biosensor fish, such as the *Tg(Hsa.CTGF:nlsmCherry)^ia49^* Yap/Taz reporter line along with the various zebrafish models of human diseases, the presence of putative crosstalk can be investigated in vivo also in specific pathological contexts. This approach offers valuable insights into the pathophysiology of various disorders, including cancer. In conclusion, our study unveils an intricate and sophisticated interplay between the Wnt/β-catenin and Hippo-Yap/Taz signaling pathways, shedding light on their roles in embryonic development and disease. Collectively, our findings contribute to a deeper understanding of the molecular mechanisms governing tissue homeostasis and organ development, paving the way for further investigations into the intricate regulatory mechanisms of cellular signaling networks and their implications for human health and disease.

## 4. Materials and Methods

### 4.1. Animals

All live animal procedures were approved by the institutional ethics committee for animal testing of the University of Padua as well as in accordance with the relevant guidelines and regulations of Italy and European Union. To inhibit pigment formation, embryos and larvae were incubated in 0.003% 1-phenyl-2-thiourea (PTU). The following fish lines were used and outcrossed with the *Tg(Hsa.CTGF:nlsmCherry)^ia49^* Yap/Taz reporter line: *Tg(7xTCF-Xla.Siam:GFP)^ia4^* Wnt/β-catenin reporter [46], *Tg(hsp70:dkk1-GFP)^w32^* heat shock inducible Dkk overexpressing line [39], apc^hu745^ mutant line [41]. The apc^hu745^ mutant carriers were genotyped by PCR amplification followed by sequencing. The genotyping primer pair was 5′-CTACCCAACTTTACCTATATCAG-3′ and 5′-GACTCTCAAAACTGTCAAGGG-3′. The PCR product was sequenced using the forward primer, and the sequence obtained was analyzed at position 33 downstream of the sequencing primer. Homozygous apc^hu745^ mutant embryos were identified and selected through their phenotype using a dissecting microscope.

### 4.2. Chemical Treatments

The following chemical compounds were used: IWR-1 (I0161, Sigma-Aldrich, S. Louis, MO, USA), XAV939 (X3004, Sigma-Aldrich, S. Louis, USA), 6-Bromoindirubin-3′-oxime (BIO) (B1686, Sigma-Aldrich, S. Louis, USA). The embryos were dechorionated and exposed to the drugs from 24 to 48 hpf in fish water with PTU.

Different concentrations of Wnt inhibitors were tested on Yap/Taz *Tg(Hsa.CTGF:nlsmCherry)^ia49^* (Appendix A) and Wnt *Tg(7xTCF-Xla.Siam:GFP)^ia4^* embryos (Appendix A).

Specifically, a preliminary test on a subset of embryos using different concentrations of the Wnt inhibitors IWR-1 (5, 10, and 15 µM) and XAV939 (2.5, 5, and 7.5 µM) (Appendix A) was conducted. After 24 h of treatment, Yap/Taz and Wnt pathway activities were assessed by measuring the changes in fluorescence levels in transgenic embryos (Appendix A). Based on the inhibitors’ efficacy and the absence of significant toxicity, the concentrations of 10 µM for IWR-1 and 5 µM for XAV939 were selected for subsequent experiments in a larger sample of embryos (Figure 2A–C). A concentration of 5 μM was used for BIO, and the corresponding volume of DMSO, in which all three chemicals were solubilized, was applied in all control treatments.

### 4.3. Fluorescence Image Acquisition and Analysis

The fluorescence was visualized using the conventional fluorescence dissecting microscope Leica (Wetzlar, Germany) M165FC with a 488 nm (for GFP) and 561 nm (for mCherry) filter, equipped with a DFC7000T digital camera (Leica, Wetzlar, Germany).

Fluorescence quantification of the acquired images was performed using Fiji software (2.9.0 version) by measuring the fluorescent signal in the whole embryo as integrated density, as described elsewhere [47]. To perform subcellular colocalization analysis, we used a plugin under ImageJ named JACoP [48].

### 4.4. Whole-Mount In Situ Hybridization (WISH) and Quantification

Whole-mount in situ hybridization was performed on embryos pre-fixed in 4% PFA in PBS overnight at 4 °C and stored in pure methanol at −20 °C as previously described [49]. mCherry probe was synthesized from the pME-nlsmCherry supplied by the Tol2 kit (Invitrogen), which was linearized and transcribed with the T7 RNA polymerase using DIG-labeled ribonucleotides.

The intensity of the in situ hybridization signal was quantified using two different methods. The first was carried out by categorizing the embryos into distinct groups based on the strength of the signal observed under the dissecting microscope. These groups were defined as exhibiting strong, medium, or weak signal intensity, allowing for a systematic assessment of the variation in signal strength between the control and experimental groups. The embryos in each subgroup were counted, and the data were plotted in a histogram. For each type of experiment, an additional quantification of mCherry signals was performed as follows: WISH acquisitions were converted to grayscale profiles and inverted to negative images. The WISH-related bright signals were quantified using the Measurement tool of the Volocity 6.0 software (Perkin Elmer, Milan, Italy) on 100 μm × 100 μm regions of interest (ROIs). Signal values were normalized to the control average value (set as 100 arbitrary units). Comparisons between control and treated populations were made using a Mann−Whitney test.

### 4.5. Alcian Blue Staining

For cartilage staining, larvae were fixed overnight in 4% PFA in PBS at 4 °C, washed in PBS, and stained overnight at room temperature in Alcian blue solution (70% EtOH, 1% HCl, and 0.1% Alcian blue). Larvae were then cleared in 3% hydrogen peroxide and 1% KOH, rinsed in 70% EtOH, and whole mounted in 85% glycerol.

### 4.6. Heat-Shock Treatments

Dkk overexpression in Yap/Taz reporter embryos was obtained by outcrossing the *Tg(Hsa.CTGF:nlsmCherry)^ia49^* reporter line to the *Tg(hsp70:dkk1-GFP)^w32^* heat shock inducible Dkk overexpressing line. The offspring was heat-shocked every 12 h, from 12 to 72 hpf, by replacing the fish water with water preheated to 40 °C and incubating the embryos in an air incubator at 37 °C for 1 h. The embryos were then sorted by GFP fluorescence, and GFP-negative siblings were used as controls.

### 4.7. Statistical Analyses

Data are presented as mean  ±  SEM. The distribution of the data was assessed using Shapiro−Wilk. Statistical comparisons between groups were performed using a two-tailed Student’s *t*-test or the non-parametric Mann−Whitney test for the quantification of the fluorescence intensity and contingency analysis for the quantification of the in situ hybridization signal. Statistical analyses were carried out with Prism 10 software version 10.3.1 (GraphPad).

All data were derived at least from three independent experiments, and the exact sample size (n) is reported in the figure legends.

## Figures and Tables

**Figure 1 ijms-25-10005-f001:**
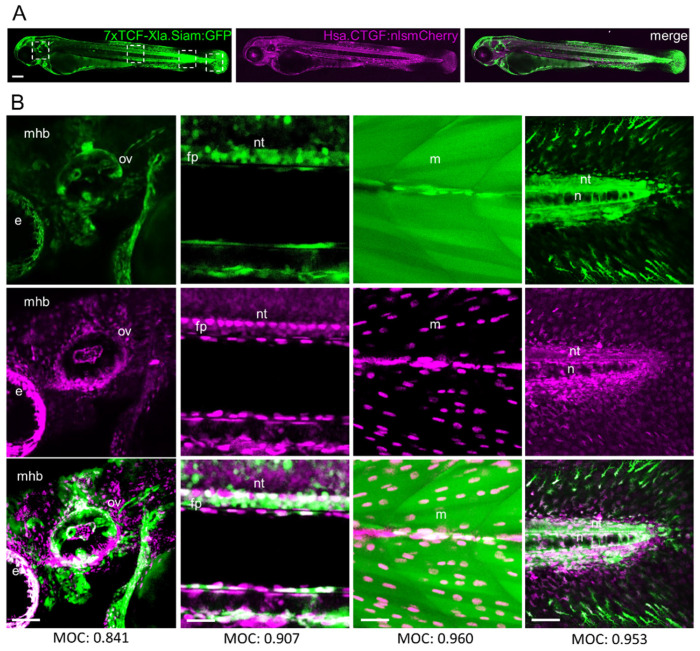
(**A**,**B**) Transgene co-expression pattern of *Tg(7xTCF-Xla.Siam:GFP)^ia4^* and *Tg(Hsa.CTGF:nlsmCherry)^ia49^* reporter lines in 48 hpf embryos. (**A**). Representative *Tg(7xTCF-Xla.Siam:GFP)^ia4^*, *Tg(Hsa.CTGF:nlsmCherry)^ia49^*, and merged images showing Wnt (green) and Yap/Taz (magenta) responsive areas. Regions selected by white dashed squares have been zoomed in (**B**). (**B**) Magnification of head, trunk, muscles and caudal region views of the maximal projection from three stacks. Abbreviation: mhb (midbrain–hindbrain boundary), ov (otic vesicle), nt (neural tube), n (notochord), fp (floorplate), m (muscle), e (eye). Mander’s overlap coefficient (MOC), Scale bar in (**A**) = 200 µm; Scale bar in (**B**) = 25 µm.

**Figure 2 ijms-25-10005-f002:**
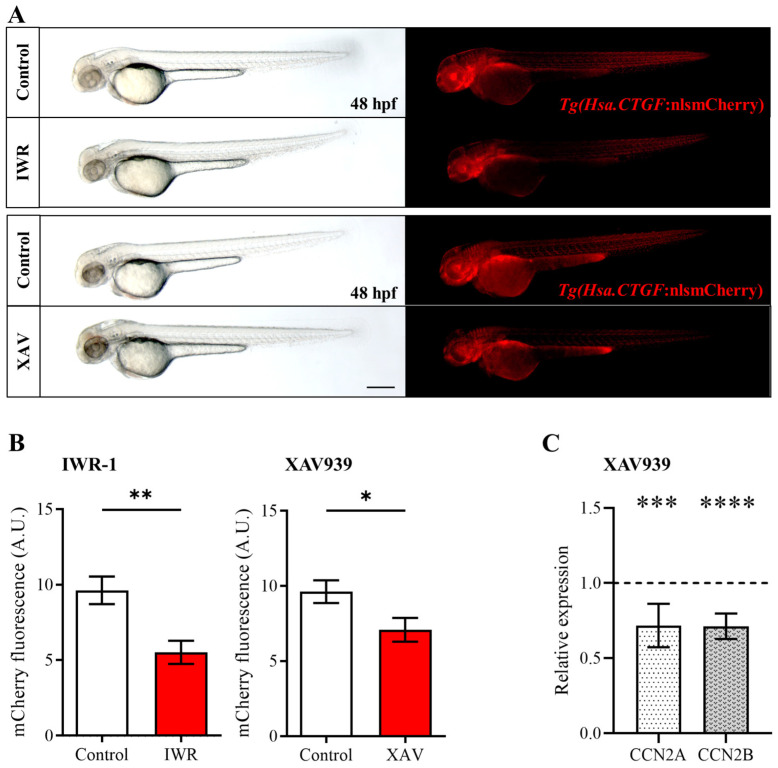
IWR-1 and XAV939-mediated β-catenin pathway inhibition reduces *Tg(Hsa.CTGF:nlsmCherry)^ia49^* reporter signal. (**A**,**B**) *Tg(Hsa.CTGF:nlsmCherry)^ia49^* embryos were exposed to IWR-1 or XAV939 for 24 h, and the fluorescent reporter expression was documented (**A**) and quantified at the level of the whole embryo (**B**) at 48 hpf. mCherry fluorescence levels (**B**) were analyzed by measuring the integrated density with FIJI software Control IWR (n = 27), IWR treated embryos (n = 28); Control XAW (n = 31), XAW treated embryos (n = 34). (**C**) Relative expression of ccn2a and ccn2b genes in zebrafish embryos after treatment with XAV939 from 24 to 48 hpf. mRNA levels were normalized to the DMSO-treated controls (dashed line). Scale bar 250 μm Data are presented as mean +/− SEM (Mann−Whitney test (**B**), Student *t*-test (**C**); * *p* < 0.05, ** *p* < 0.01, *** *p* < 0.001, **** *p* < 0.0001).

**Figure 3 ijms-25-10005-f003:**
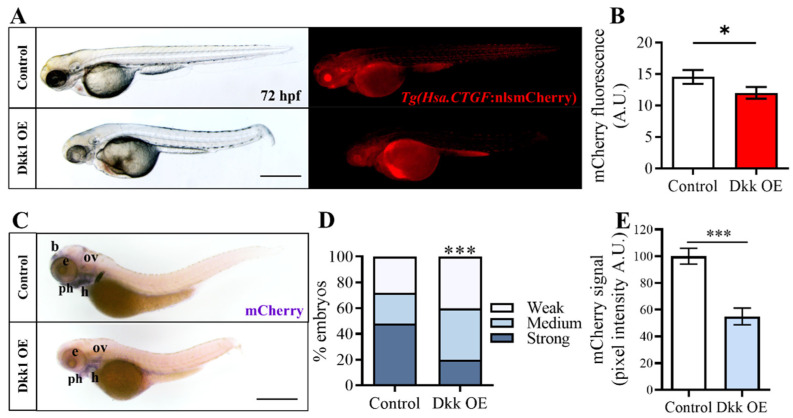
Wnt/β-catenin pathway inhibition through Dkk1 overexpression reduces *Tg(Hsa.CTGF:nlsmCherry)^ia49^* transgene expression. (**A**) *Tg(Hsa.CTGF:nlsmCherry)^ia49^* fish were outcrossed with the *Tg(hsp70:dkk1-GFP)^w32^* line, and the offspring was heat shocked to induce the overexpression of the Wnt antagonist Dkk1 (Dkk1 OE). Representative images of a double transgenic embryo (Dkk1 OE) at 72 hpf after Dkk1 overexpression compared to WT control siblings (control). (**B**) mCherry quantification on the entire embryo showed only a modest but significant decrease of fluorescence intensity in *Tg(hsp70:dkk1-GFP)^w32^* larvae after Dkk1 activation; Control (n = 84), Dkk OE (n = 87). (**C**) In-situ hybridization performed with an antisense probe against mCherry transcript revealed a significant reduction of mCherry mRNA levels after Dkk1 overexpression, compared to the control siblings. The decrease in mCherry mRNA levels is evident in the brain (b), eyes (e), otic vesicle (ov), heart (h), and pharyngeal arches (ph). (**D**,**E**) The variation in the in situ hybridization signal intensity was quantified using two independent systems. In (**D**), the analysis was performed by manually categorizing the embryos of the control and experimental samples (Dkk1 OE) into distinct groups based on the observed signal strength (strong, medium, and weak) at the whole embryo level. In (**E**), the mCherry signals in each sample (head region) were quantified using the Volocity 6.0 software. Control (n = 321), Dkk OE (n = 288). Scale bar 500 μm. In (**B**,**E**), data are presented as mean +/− SEM (Mann−Whitney test); * *p* < 0.05, *** *p* < 0.001).

**Figure 4 ijms-25-10005-f004:**
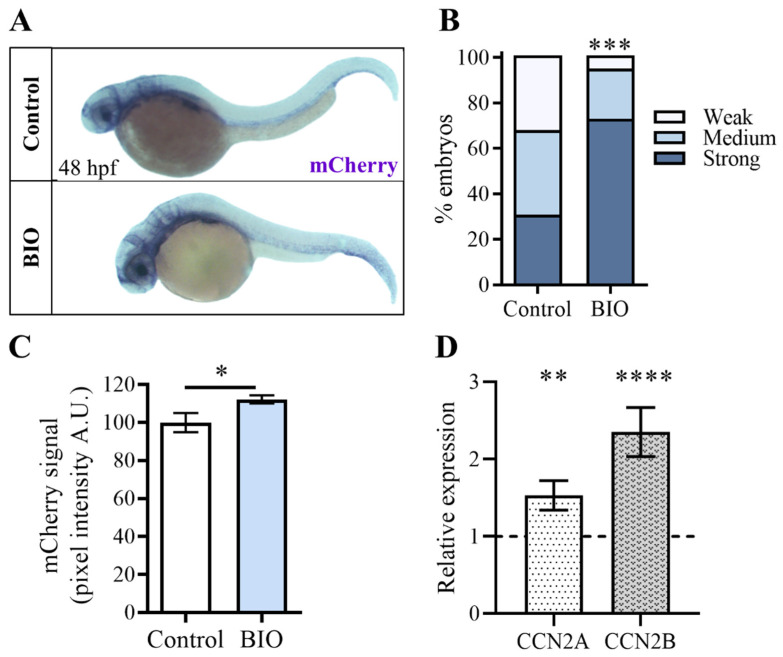
Pharmacological inhibition of β-catenin kinase GSK3 increases *Tg(Hsa.CTGF:nlsmCherry)^ia49^* reporter signal. (**A**) *Tg(Hsa.CTGF:nlsmCherry)^ia49^* embryos were exposed to BIO from 24 to 48 hpf, and after fixation, the mCherry transgene expression was assessed by in situ hybridization. (**B**,**C**) The embryos treated with BIO showed higher levels of mRNA mCherry transgene expression compared to their untreated control siblings. In the in situ hybridization experiments, the signal intensity variation was assessed using two different methods. In (**B**), the analysis was performed by grouping the embryos into three distinct classes based on the observed signal strength at the whole embryo level. In (**C**), the mCherry signals were quantified by measuring the pixel intensity in the head region, using Volocity 6.0 software. Control (n = 52), BIO (n = 51). (**D**) In 48 hpf zebrafish embryos, the expression of *ccn2a* and *ccn2b* genes was strongly up-regulated after treatment with BIO for 24 h. mRNA levels were normalized to the DMSO-treated controls (dashed line). Scale bar: 250 μm. In (**C**,**D**), data are presented as mean +/− SEM (Mann−Whitney test in (**C**) and Student *t*-test in (**D**); * *p* < 0.05, ** *p* < 0.01, *** *p* < 0.001, **** *p* < 0.0001).

**Figure 5 ijms-25-10005-f005:**
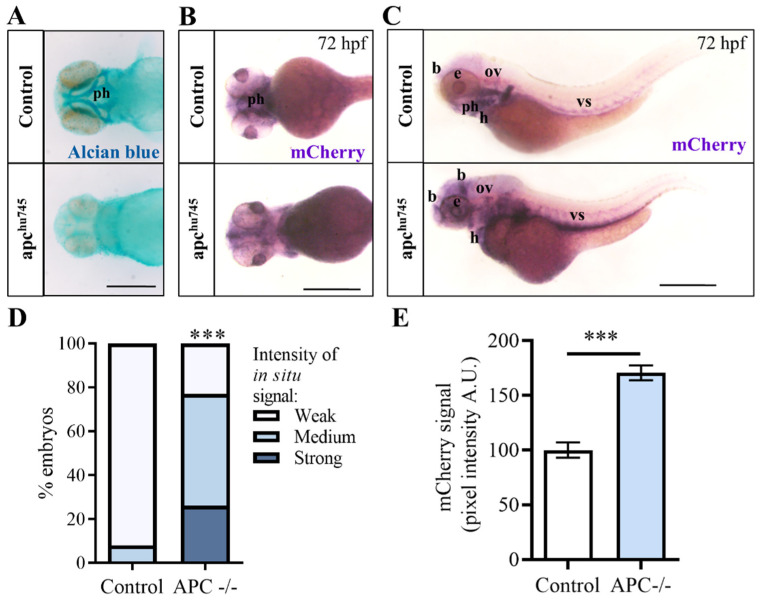
*Tg(Hsa.CTGF:nlsmCherry)^ia49^* reporter activity is increased in apc mutant background at 72 hpf. (**A**) apc^hu745^ mutants fail to develop the cartilage structure, as evidenced by the alcian blue staining. (**B**) The reporter signal, labeling the pharyngeal arches (ph), disappears in apc^hu745^ mutant background due to the lack of these structures, as shown by the in situ hybridization against mCherry mRNA as well. (**C**) In situ hybridization for mCherry displaying the general upregulation of Yap/Taz reporter activity in apc^hu745^ mutant background. The increase in mCherry mRNA levels is evident in the brain (b), eyes (e), otic vesicle (ov), heart (h), and vascular system (vs) along the yolk extension. (**D**,**E**) Quantification of the in-situ hybridization experiments was performed (**D**) by grouping the embryos into three distinct classes according to the observed signal strength at the whole-embryo level, or (**E**) by measuring the pixel intensity in the head region of each sample using Volocity 6.0 software. Control (n = 48), APC^−/−^ (n = 39). In (**E**), data are presented as mean +/− SEM (Mann−Whitney test). Scale bar: 500 μm; *** *p* < 0.001.

## Data Availability

The data presented in this study are available in the article and in the Appendix A.

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
