# Peer review of "Wnt/β-Catenin Signaling Regulates Yap/Taz Activity during Embryonic Development in Zebrafish"

_ijms, 2024, doi:10.3390/ijms251810005_

Round 1
Reviewer 1 Report
Comments and Suggestions for Authors
Hippo-YAP/TAZ and Wnt/β-catenin signaling pathways are crucial to development and tissue homeostasis. Interactions between these two signaling pathways had been found in embryogenesis and disease development. In this study the authors analyzed the ability of the Wnt/β-catenin pathway to regulate YAP/TAZ activity during zebrafish development at the whole organism level, they found that Hippo-YAP/TAZ and Wnt/β-catenin reporter zebrafish lines shared spatiotemporal expression profiles, and modulation of Wnt/β-catenin pathway by both chemicals or genetic perturbation resulted in significant regulation of the YAP/TAZ activity. These results provided demonstrations on Wnt/β-catenin-mediated regulation of Yap/Taz activity in a living vertebrate model. But there are several comments for the authors addressing.
1. The authors took the signal of the whole embryos for evaluation of the reporter activity or expression level of certain genes, but the density of the signal in different organs or tissues are not completely the same, it might be more convincing if just took selected tissues that had been proved to be regulated by the two signalings.
2. In Figure 1, the reporter expression pattern of Tg(7xTCF-Xla.Siam:GFP)ia4 and Tg(Hsa.CTGF:nlsmCherry)ia49 in otic vesicle, notochord and muscle looks not overlapping exactly, so the authors should not state that “Wnt/β-catenin and Hippo-Yap/Taz pathways are characterized by an overlapping spatial expression during zebrafish development”.
Reviewer 2 Report
Comments and Suggestions for Authors
Wnt/β-Catenin signaling and Yap/Taz hippo signaling were reported to cross-talk with each other in embryonic and tumorigenic tissues. The authors took advantage of the zebrafish Yaz-Yap reporter fish line, Tg(Hsa.CTGF:nlsmCherry), showing evidence that wnt signaling may regulate Yap/Taz activity in zebrafish. Overall, the topic is interesting. However, some experimental results are quite preliminary and data quality can be improved. Thus, the conclusion is too strong, given the presented experimental evidence.
Major issues:
1. Some results quality needs to be improved. For example, the In situ hybridization images has a higher purple background, which masked signals.
Figure 1. The co-expression data are too preliminary. Single-cell resolution imaging and quantification should be considered to demonstrate co-expression. With the current presented data, it is hard to know whether the GFP and mCherry are in the same cells or different cells within the same organs.
Figure 2. It may be hard to choose the right dosage of inhibitors. A single dose of data is not enough to draw a conclusion. It is better to include a dosage response. This also helps to exclude the decrease of mCherry is not caused by toxicity.
Figure 4. mCherry should be also quantified by fluorescence or RT-PCR. In in situ hybridization is not a suitable quantitative method. Also, the trunk and head mRNA levels are not consistent in Panel A. There is an increase in the head but there seems to be a decrease in the trunk. This difference should be reported and discussed.
Figure 5. Similar to Figure 4, the mCherry in situ is not a good approach for quantification. Fluorescence or RT-PCR should be given.
2. It is well known that zebrafish have duplicated wnt genes and two ctgf genes. So, there will be a redundancy in the signal cross-talks between the two pathways. The Tg(Hsa.CTGF:nlsmCherry) fish may not faithfully report Yap/Taz hippo signaling. This limitation should be discussed.
Minor issues:
Line 130-136: These sentences are the next step, which is to investigate the regulation between two pathways by interventions. They should be moved to the beginning of section 2.2.
Line 148-150 (section 2.2): How is the mcherry fluorescent are measured and normalized? Whole embryos or certain tissues or organs?
Line 222 (figure 4): The control in Panel A is labeled as a 48 hpf embryo, while the BIO-treated embryo looks like a younger 24hpf embryo.
Figures 3-5: Anatomic structures should be labeled on the figure to make them more readable.
Author Response
Please see the attachment for reply to the referee n. 2

Round 2
Reviewer 2 Report
Comments and Suggestions for Authors
In this revision, most of the concerns were addressed, and the overall quality of the manuscript improved. However, some issues still remain. Based on the data in Figure 1, the mCherry-positive cells do not overlap with the GFP-positive cell. They are almost exclusively. So, the co-expression conclusion does not stand. This needs to be addressed, and the overall interpretation also needs to be changed accordingly. For the data in Figure 2, the single dosage is still not enough, although the chosen dosage is based on the literature. The genetic backgrounds of zebrafish from different labs are very different. So, the dosage from the literature may serve as a starting point, but not as a standard. The dosage effect is stronger evidence than a single one, which cannot exclude the toxicity effect.
Comments on the Quality of English LanguageThe quality of English is OK.
Round 3
Reviewer 2 Report
Comments and Suggestions for Authors
The supplementary figures are not mentioned in the current version of manuscript. Please add them in the suitable result sections. Alternatively, the data pf these two new supplementary figures can be integrated into the main figure 1 and figure 2.
Comments on the Quality of English LanguageNo concern.
Author Response
Pleas see the attachment
